# Effects of Functional and Nutraceutical Foods in the Context of the Mediterranean Diet in Patients Diagnosed with Breast Cancer

**DOI:** 10.3390/antiox12101845

**Published:** 2023-10-11

**Authors:** Giovanna Flore, Andrea Deledda, Mauro Lombardo, Andrea Armani, Fernanda Velluzzi

**Affiliations:** 1Obesity Unit, Department of Medical Sciences and Public Health, University of Cagliari, 09124 Cagliari, Italy; gflore@unica.it (G.F.); andredele@tiscali.it (A.D.); fernandavelluzzi@gmail.com (F.V.); 2Department of Human Sciences and Promotion of the Quality of Life, San Raffaele Roma Open University, 00166 Rome, Italy; mauro.lombardo@uniroma5.it; 3Laboratory of Cardiovascular Endocrinology, Istituto di Ricovero e Cura a Carattere Scientifico (IRCCS) San Raffaele, 00166 Rome, Italy

**Keywords:** breast cancer, functional nutrition, Mediterranean diet, antioxidants, obesity, epigenetic, nutrigenomics, quality of life, therapy side effects

## Abstract

Several studies report that breast cancer survivors (BCS) tend to have a poor diet, as fruit, vegetable, and legume consumption is often reduced, resulting in a decreased intake of nutraceuticals. Moreover, weight gain has been commonly described among BCS during treatment, increasing recurrence rate and mortality. Improving lifestyle and nutrition after the diagnosis of BC may have important benefits on patients’ general health and on specific clinical outcomes. The Mediterranean diet (MD), known for its multiple beneficial effects on health, can be considered a nutritional pool comprising several nutraceuticals: bioactive compounds and foods with anti-inflammatory and antioxidant effects. Recent scientific advances have led to the identification of nutraceuticals that could amplify the benefits of the MD and favorably influence gene expression in these patients. Nutraceuticals could have beneficial effects in the postdiagnostic phase of BC, including helping to mitigate the adverse effects of chemotherapy and radiotherapy. Moreover, the MD could be a valid and easy-to-follow option for managing excess weight. The aim of this narrative review is to evaluate the recent scientific literature on the possible beneficial effects of consuming functional and nutraceutical foods in the framework of MD in BCS.

## 1. Introduction

According to the International Agency for Research on Cancer (IARC), “in 2020 breast cancer became the most commonly diagnosed cancer type in the world; there were more than 2.26 million new cases of breast cancer and almost 685,000 deaths from breast cancer worldwide” [1]. Breast cancer (BC) is the leading cause of cancer-related death in women, despite the discrepancies in case-fatality rate among different countries, with a lower mortality in high-income countries, linked to the widespread programs for diagnosis and prevention [2].

It is possible to classify BC based on the molecular receptor status. Three receptors have been characterized: human epidermal growth factor receptor-2 (HER-2), estrogen receptor (ER), and progesterone receptor (PR). The subtypes of BC are luminal A and B (respectively, ER/PR-positive/HER-2-negative and ER-positive, PR-positive or negative, and HER-2-positive); HER-2-enriched; claudin-low (ER/PR-negative, HER-2-positive, and low claudin expression); triple-negative, including the basal-like phenotype (ER/PR/HER-2-negative); normal breast-like (ER/PR-positive or negative and HER-2-negative). Luminal mammary tumors (luminal A/B), which are characterized by positive expression of ER and/or PR, are the most common subtypes. Luminal B is considered more aggressive than luminal A, and along with the triple-negative subtype is associated with a worse prognosis [3].

With regard to the etiopathogenesis, the development of BC seems to be mainly linked to environmental, reproductive and lifestyle factors [4,5], while less than 10% of BC may be attributed to inherited genetic mutations [6]. Indeed, the incidence of BC is higher in Europe and North America than in Asian countries, probably due to the different lifestyle [7,8]. Age, family history, early menstrual cycle or late menopause are non-modifiable risk factors [9]. Absence or late pregnancy [10], hormone replacement therapy [11], alcohol consumption [12], failure of breastfeeding [13,14], physical activity level [15], and overweight and obesity [16] are modifiable risk factors [17]. In particular, obesity seems to play a crucial role, and it has been shown that weight loss, including after bariatric surgery, significantly reduces BC risk [18,19,20].

On the other hand, the risk factors for recurrence and mortality in BC survivors (BCS) are less clear, and several factors should be considered, among which are the length and timing of exposure to each factor, specifically before or after diagnosis. Table 1 summarizes the BC risk and protective factors, according to the degree of evidence in the latest World Cancer Research Fund (WCRF) reports for healthy subjects (before BC diagnosis), among premenopausal and postmenopausal subjects and BCS [21].

At the same time, while well-studied guidelines exist for BC prevention [22], there remains a need to identify a lifestyle that can be effective in improving the long-term health of BCS, and more specifically, in reducing the recurrence rate and mortality.

Despite the few and non-conclusive data, some evidence highlights an association between the adherence to a Western or unhealthy diet, both before and after the diagnosis of BC, and overall and non-BC mortality [23,24,25]. Moreover, unhealthy diet and sedentary behavior are among the main factors promoting obesity [26], which in turn is a risk factor for BC recurrence and mortality in BCS [27]. In contrast, a reduction in overall mortality was found to be associated with a higher adherence to a prudent or healthy diet [28]. In addition, other studies suggest that a diet low in fat, especially saturated fat, and rich in vegetables and fruits may be protective toward recurrences in BCS [28,29,30]. Furthermore, such a diet has been demonstrated to positively modulate the gut microbiota (GM) [31]. The composition and functionality of the GM have been linked to BC in several clinical studies [32]. (Figure 1)

In this context, the Mediterranean diet (MD), a plant-based diet, which is considered one of the healthiest nutritional models [33], has potential to help in the postdiagnosis phase of BC, due in part to the presence of several nutraceuticals: bioactive compounds and food components that, in addition to nutritional properties, have well-established antioxidant, anti-inflammatory, and anticancer effects [34].

This narrative review, aimed at investigating how the MD, rich in functional foods and nutraceuticals, can be a potential adjuvant to the treatment of patients with BC. In particular, we focused on which foods, spices, and aromas typical of the MD, if frequently consumed, can exert antioxidant, anti-inflammatory, and anticancer effects.

Our hypothesis stems from the consideration that nutraceuticals, when consumed in their natural state within foods, can exhibit synergistic effects. These effects can occur both among various components within a single food and between different foods consumed in the same meal, potentially increasing the bioavailability and efficacy of nutraceuticals.

Finally, given the well-established negative role of overweight and obesity among BCS, we also evaluated the effects of the MD and its compounds on weight control.

## 2. Methods

For this review, a PubMed and Cochrane search was performed using the following key words: “Mediterranean diet and Breast Cancer”; or “Nutraceuticals and Breast Cancer”; or “Functional foods and Breast Cancer”; or “Functional foods and Mediterranean Diet” or “Nutraceuticals and Mediterranean diet”; or “Breast Cancer and Nutrition” or “Food Sinergy”. We selected articles in English published between 2012 and February 2022. Additional articles were obtained from selected articles’ bibliographies or by specific search of singular nutraceuticals or for discussion purposes (Figure 2).

Each abstract was screened independently by two researchers (GF and AD), with disagreements arbitrated by a third researcher (FV) where necessary.

Data Analysis: Due to heterogeneity of the included studies in terms of study design, intervention details, and outcomes, a meta-analysis was not feasible. Instead, the results of the studies were analyzed qualitatively, and the findings are presented in a narrative form.

Ethical Approval: This study did not require ethical approval as it is a narrative review of previously published studies.

## 3. Excess Body Fat and Breast Cancer

Several studies have demonstrated a positive association between high adiposity, as indicated by either a body mass index (BMI) ≥ 25 Kg/m^2^ (overweight) or ≥30 Kg/m^2^ (obesity), as well as elevated body fat levels, and risk of BC in the postmenopausal phase [35,36,37]. This association has not been demonstrated in the premenopausal phase [38].

Adipose tissue is an “endocrine organ”, producing factors that influence systemic energy metabolism, endocrine function, immune responses, and steroid hormone metabolism [39]. Estrogens are particularly important, especially in estrogen receptor positive (ER+) tumors [40], and aromatase, the enzyme responsible for the conversion of androgens into estrogens, has a key role in their biosynthesis [41].

In premenopausal women, estrogens are mainly synthesized by the ovaries, and specifically by the granulosa cells, through the aromatization of the androgens produced by the theca cells. The biosynthesis of estrogens in the ovaries, predominantly Estradiol (E2), is controlled by the luteinizing hormone and the follicle stimulating hormone stimulation [42].

Aromatase is also expressed in other tissues, particularly in visceral and subcutaneous adipose tissue [41].

In postmenopausal women, estrogens are mainly produced in the adipose tissue by the local aromatization of circulating androgen precursors (Figure 3). The predominant estrogen in postmenopause is Estrone (E1), a weak compound which is derived from the conversion of circulating androstenedione, and that can be transformed into the most active E2 by the action of 17b-hydroxysteroid dehydrogenase. More specifically, E1 shows higher concentrations in visceral than in subcutaneous adipose tissue; however, both E1 and E2 concentrations increase with the increase in visceral adiposity [43], linking the multiple alterations associated with visceral obesity, such as metabolic changes, the chronic inflammatory status or immunological dysfunction to the risk of postmenopausal BC [44,45]. The menopausal transition is characterized by an increase in body fat mass, with a shift in the body fat distribution from subcutaneous to visceral depots, as well as by a decrease in body lean mass, linked to the physiological decline of circulating E2 levels [46] and increased Testosterone/E2 ratio [47]. The different fat distribution in pre- and postmenopause may be attributed to the different expression of the estrogen receptors, α and β, in adipose cells [48], and overall, the postmenopausal hormonal changes result in obesogenic alterations involving lipoprotein lipase activity, eating pattern, food intake, and energy expenditure [49].

High levels of adiposity may be responsible for the increased risk of postmenopausal BC by increasing serum E2 concentration and estrogen levels in the breast [50], which may create local conditions that promote estrogen-related stimulation of mammary cancer cells [51]. A high level of aromatase activity may also be exhibited by the spiked fibroblast layer of adipose tissue surrounding breast pre-oncogenic or oncogenic lesions [41].

Moreover, the excess body fat, especially visceral fat, is associated with the secretion of several adipocytokines, such as leptin, Monocytic Chemiotactic Protein 1 (MCP-1), Tumoral Necrosis Factor α (TNF α), and interleukin (IL)-6, IL-8, and IL-1β, as well as cyclooxygenase-2 (COX-2) [52]. Among these, TNF α and IL-6 can amplify the action of aromatase in the adipose tissue, and in turn can increase the biosynthesis of estrogens [53], while the hyperinsulinemia associated with insulin resistance in visceral obesity reduces the hepatic production of sex hormone-binding globulin, increasing the estrogen’s bioavailability [54].

Furthermore, it has been observed that women with obesity have higher levels of COX-2 and prostaglandin E2 (PGE2) in their mammary tissue compared to lean women [55]. Several studies have revealed that chronic inflammation increases the risk of a variety of cancers [56], including BC, both in pre- and postmenopausal women [51,57]. In particular, chronic inflammation, along with the metabolic alterations associated with visceral adiposity, could increase the risk of premenopausal ER-negative BC [58].

Finally, the higher levels of inflammation with ageing may be responsible for an increase of visceral adipose tissue through the reduction of catecholamine-mediated lipolysis [59].

Besides the role of excess weight as a risk factor for BC, weight gain has been commonly observed within the first year following the diagnosis of BC, and it tends to persist over time [60].

Among BCS, the presence of excess body fat, especially visceral fat and the linked inflammatory environment, has been associated with an increased risk of recurrence and mortality. The risk is higher when the increase exceeds 10% of weight at diagnosis [27,59,61,62]. Although the mechanisms underlying this association are not fully understood, a possible explanation is that the metabolic alterations and chronic inflammation characteristic of visceral obesity may interfere with the cellular pathways that regulate the cell cycle and apoptosis. This, in turn, may affect tumor progression [63].

Given that a significant percentage of women experience weight gain during and after BC treatment [60], a proper weight management through a healthy diet and adequate physical activity can be an effective strategy in the control of BC. Adopting a healthier lifestyle has shown to be beneficial in preventing BC, in part due to the positive effects in preventing metabolic alterations [64]. Moreover, it plays a crucial role after a cancer diagnosis, reducing both the risk of recurrence and overall mortality [65,66]. On this issue, some studies evaluated the effects of different nutritional interventions on excess weight in BCS, but there is currently insufficient evidence to determine the optimal approach [63].

Healthy diets are based on a high consumption of vegetables, fruits, whole grains, and legumes, along with a low consumption of animal-based foods, particularly those rich in saturated fat and processed meats [67]. Among the various diets characterized by this nutritional pattern, we focused in particular on the MD, which also incorporates other aspects of lifestyle that are essential for body weight management, such as regular physical activity and adequate rest [27].

## 4. Mediterranean Diet and Breast Cancer

Providing a unique definition of MD is not easy [68]. The MD should be understood as the typical dietary style of countries bordering the Mediterranean Sea, characterized by the consumption of certain foods, with attention to their origin and cooking methods, combined with regular physical activity, adequate rest, and proper conviviality [33].

Although the MD includes different eating styles based on the geography, culture, and local traditions of the various Mediterranean countries, there are some common features, such as the daily consumption of cereals, especially whole grain cereals; other fiber-rich unrefined grains; and seasonal fresh fruits and vegetables of several colors and textures, rich in micronutrients, fiber, and phytochemicals. Olive oil is the main source of dietary fat; olives, nuts, and seeds are also included. Protein intake is preferably represented by legumes, fish, or seafood, as well as a moderate consumption of eggs and white meat, while red or processed meat could be included 1–2 times per month in small portions. Calcium intake is provided by the daily consumption of dairy products, especially low-fat yoghurt or, less frequently, small portions of cheese. Good hydration is ensured by a daily intake of at least 1.5/2.0 L of water, and it is also contemplated a moderate intake of wine, preferably red, and always with meals [67,69,70].

Based on current evidence, MD is a recommended dietary approach in the prevention of various noncommunicable diseases (NCDs), including cardiovascular disease (CVD), type 2 diabetes, and some cancers [33].

With regard to BC, a high adherence to a Mediterranean dietary pattern has been shown to be beneficial both in the prevention and in the reduction of mortality in BCS [71]. On the contrary, Western diets (WD), abundant in total meat, red and processed meat, or high-glycemic-index (GI) foods, appear to be associated with a greater risk of BC [72].

Moreover, MD has been shown to be negatively associated with weight gain [73,74], particularly with abdominal and visceral adiposity [75,76], and a higher adherence to this nutritional model is likely associated to weight maintenance [77,78]. Therefore, due to the role of excess weight and visceral fat in the development and in clinical outcomes of BC, the MD could be an optimal dietary choice. The increased likelihood of maintaining weight, which is a crucial issue in the management of overweight and obesity [79], should be considered an additional important reason to recommend this diet in BCS. Indeed, there are limited data regarding long-term weight maintenance among these patients [80].

The favorable effects of the MD have been attributed to its characteristic ingredients, which are known for their lipid-lowering, insulin-sensitizing, antioxidant, anti-inflammatory, and antithrombotic properties [81]. The MD is rich in beneficial substances, such as minerals, vitamins, lecithin, sulfides, salicylates, phytoestrogens, phytosterols, polyphenols, and glucosinolates, found in fruits and vegetables. These compounds provide a protective effect by acting as antioxidant, deactivating carcinogens, preventing spontaneous mutations and oxidative DNA damage related to metabolic processes [82].

Classic examples are the phenolic compounds present in EVOO (extra virgin olive oil), which are useful in reducing LDL and its oxidation, oxidative DNA damage, and the production of proinflammatory cytokines; increasing insulin sensitivity through cell membrane modification; and improving endothelial function through increased bioavailability of vasodilating agents. Additional benefits of the MD are attributable to fiber, phytosterols, and polyphenols such as resveratrol, MUFAs (monounsaturated fatty acids), and PUFAs (polyunsaturated fatty acids), vitamins, and minerals [34,71].

Foods rich in micronutrients can exhibit nutraceutical properties. Nutraceuticals are indicated as “any substance that may be considered a food or part of a food, and provides medical or health benefits, including the prevention and treatment of disease” [83] (Figure 4). The MD is a food model particularly rich in nutraceutical foods. A cornerstone of this dietary pattern is the aforementioned EVOO, with a high content of MUFAs. Other compounds, such as squalene, triterpenes, pigments, tocopherols, or phenolic compounds, although present in small amounts, possess antioxidant, anti-inflammatory, and anticancer properties, or act as regulators of the GM [31,84]. The MD is also rich in senolytic molecules, such as oleuropein and (-)-epigallocatechin gallate (EGCG), which are natural or artificial compounds able to act against senescent cells. These cells are known for their disrupting action towards metabolism and are associated with a higher risk of cancer [85].

In this review, we will examine the most extensively studied micronutrients and phytonutrients with anticancer nutraceutical properties that are components of the MD.

### 4.1. Phenols

In recent decades, several studies have indicated the chemosensitizing effects of food phenols in patients with BC. However, these compounds have a scarce bioavailability in their native form because after ingestion, phenolic compounds may undergo modification by the GM or enterocytes to generate different molecules. These molecules can be found in circulation for several days and can mediate the biological effects of their parental compound [86]. Transformation of dietary polyphenols may affect their biological activity [87,88,89]. For instance, some metabolites of the GM, derived from ellagic acid and ellagitannins, have shown antiproliferative action on cancer cells [90].

Cell-based experiments have shown that curcumin treatment was able to reduce the levels of the antiapoptotic factor Bcl-2 in cultures of BC cells [91]. Another study has shown that treatment with green tea polyphenols and sulforaphane results in chromatin changes and induces activation of tumor suppressor genes such as p21(CIP1/WAF1) and Klotho [92]. In addition, mammary tumors carrying BRCA1 mutations display low levels of SIRT1 and increased levels of Survivin. BRCA1 has been shown to increase SIRT1 expression, which in turn inhibits Survivin, whose function is expected to favor tumor growth. Increased activity of SIRT1 by resveratrol reduces Survivin expression, resulting in inhibitory effects on BRCA1 mutant cancer cell expansion. These data suggest that resveratrol represents a promising therapy against breast tumors associated with BRCA1 mutations [93]. Antineoplastic properties have also been shown by ferulic acid, a polyphenol isolated from *Ferula foetida*, which is able to reduce the viability of BC cells MDA-MB-231, while also reducing the metastatic potential by counteracting the epithelial–mesenchymal transition [94].

Evidence for the efficacy of polyphenols in in vivo models of breast tumor has been provided by Luo et al. [95] who inoculated 4T1 BC cells in Balb/c mice cotreated with the antitumor Paclitaxel and the green tea polyphenol EGCG. The study showed that EGCG promoted apoptosis of cancer cells and inhibited tumor growth. The antitumor effects of EGCG were also confirmed by Thangapzham et al. [96] in nude mice inoculated with MDA-MB-231 cells. Treatment of these mice with EGCG resulted in inhibition of tumor proliferation via induction of apoptosis in cancer cells. Decreased expansion of MDA-MB-231 tumors, through increased apoptosis and reduced angiogenesis, was also observed in nude mice upon treatment with resveratrol, confirming the potential anticancer properties of this polyphenol, also in vivo [97].

In general, diets rich in phenols, (flavonols, anthocyanins, flavanones, lignans, flavan-3-ols, resveratrol, curcumin, and hydrolyzable tannins), are associated with reduced cancer growth, incidence, and metastasis, as well as a longer period of cancer latency, and linked to a higher survival rate [86].

As regards isoflavones, which are contained in a wide range of food sources, such as soy, asparagus, and borlotti beans, different studies have shown inhibitory or proliferative effects on BC cells, depending on experimental models and conditions. Interestingly, the phase of life in which consumption of isoflavones begins influences the risk of BC, which may be lower when soy is consumed in the diet from the early years of life. This could explain the greater chemoprotective effect of soy-based products in Asian women, who consume such foods throughout life, while Western women often introduce soy-based foods only in adulthood. Notably, modulation of the risk of cancer recurrence by isoflavones might be also affected by genetic polymorphisms, in particular in genes involved in this disease, i.e., cytochrome P450 1B1, the oncogene MDM2, and p53 [98,99].

Besides the more specific antineoplastic effects described above, some evidence supports a beneficial effect of dietary polyphenols in reducing excess body fat. Several mechanisms have been proposed to explain their benefits, such as the inhibition of digestive enzymes promoting obesity, the modulation of neuropeptides involved in eating behavior, the stimulation of energy expenditure or mitochondrial biogenesis, and favorable changes in gut microbiota profile, suggesting a potential role of these compounds in the treatment of obesity [100]. Given the commonly observed weight gain in BCS, and the strong impact of obesity on the clinical outcomes of BC, a diet rich in polyphenols, such as the MD, could provide a valid support for these patients, also due to its antiobesity action. However, defining the exact level of a bioactive substance in food is difficult, and further research is needed to establish the optimal intake of dietary polyphenols.

#### 4.1.1. Polyphenols in Extra Virgin Olive Oil

Extra virgin olive oil (EVOO) is mainly composed of monounsaturated acids (oleic acid) and, to a lesser extent, of acids with eighteen carbon atoms omega 6 and 3, linoleic acid (LA), and α-linolenic acid (ALA), and a low content of saturated fatty acids. Tocopherols are also present, as well as phenolic and alkaline acids, flavonoids, secoiridoids, lignans, and hydroxysichromans. The phenolic portion exhibits an antioxidant action, which is very useful for the long shelf life of EVOO. The consumption of EVOO is associated with a low incidence of atherosclerosis, skin diseases, inflammatory diseases, autoimmune diseases, diabetes, ageing, and tumors. Oleocanthal polyphenol, which is responsible for the pungent taste sensation of EVOO, has inhibitory actions on inflammatory diseases similar to non-steroid anti-inflammatory drugs (NSAID) [101,102]. In vivo and in vitro studies show that EVOO and/or its compounds can influence the promotion, initiation, and progression of carcinogenic processes through multiple and various direct or indirect mechanisms, which are able to influence different signaling pathways. They are, in fact, capable of inhibiting proliferation, inducing apoptosis, arresting the cell cycle, and reducing inflammation, immune evasion, migration, and angiogenesis [103]. Finally, polyphenols in olive oil appear to have lipid-lowering, insulin-sensitizing, and prebiotic effects, thus contributing to metabolic health [104,105].

#### 4.1.2. Quercetin

Quercetin is a flavanol mainly present in some vegetables such as red onion, capers, lettuce, cruciferous vegetables, celery, tomatoes, asparagus, and shallots [85]. It is also present in some fruits such as berries, green tea, black grapes, pomegranates, citrus fruits, apples, pistachios, nuts, and animal products such as propolis. Overall, the quercetin content in various foods ranges from 1.8 mg/100 g in foods like red onions or cruciferous vegetables to 3.5 mg/100 g in berries and 4.7 mg/100 g in apples [106].

The biosynthesis of quercetin occurs due to the action of the sun, after which it is stored in the peel of fruit and in the leaves [107].

The antioxidant activity of quercetin is related to the presence of a phenolic group in its chemical formula. It also expresses anti-inflammatory, antimicrobial and antidiabetic effects [85].

In the field of oncology, it has been observed that quercetin, even at low doses, can exhibit chemopreventive effects due to its antioxidant action, while at high doses, it exerts chemotherapeutic effects, acting as a pro-oxidant. In mouse models, 1 mg/Kg for 15/30 days reduces breast tumor volume. Six doses of 1 mg/Kg given orally every three days reduce the tumor volume of breast adenocarcinoma in mice. Meanwhile, 5 mg/Kg administered intraperitoneally twice daily for a month reduces the tumor size of breast cancer in mouse models [108].

Anticancer effects of quercetin are based on its ability to reduce the proliferation of cancer cells, inducing apoptosis, arresting the cellular cycle and inhibiting mitotic processes. In particular, in cultures of MCF-7 human BC cells, quercetin has been shown to induce apoptosis and necroptosis by modulation of Bcl-2 and Bax expression levels [109]. Another study has investigated the effects of quercetin in MCF-7 BC cells treated with the chemotherapeutic drug 5-fluorouracil (5-FU) [110]. Quercetin has been shown to enhance the anticancer properties of 5-FU by inducing apoptosis through increased levels of Bax, p53, and caspase-9 and, in parallel, reducing Bcl2 expression. These data suggest that quercetin is able to enhance the sensitivity of BC cells to 5-FU [110].

Other studies have shown that quercetin can be used as adjuvant treatment in combination with the antimitotic compound docetaxel [111]. In fact, at the molecular level, upregulation of the transcription factor Lef1 leads tumor cells to become drug-resistant. Cell culture experiments displayed that the reduced expression of Lef1 by treatment with quercetin enhances the sensitivity of BC cells to docetaxel [111].

In human MDA-MB-231 BC cell cultures, quercetin treatment was able to counteract the metastatic phenotype and epithelial–mesenchymal transition by reducing the levels of the transcription factors Snail and Slug and decreasing IGF-1 production. In order to investigate the impact of quercetin on IGF1-induced invasion and metastasis in vivo, MDA-MB-231 cells were implanted into SCID mice treated with this flavanol, which was able to reduce the pulmonary nodules and lung metastases formation in these animals. In tumor tissues, quercetin induced a reduced expression of mesenchymal markers (i.e., Snail, Slug, fibronectin), in parallel, increasing the levels of the epithelial markers keratin 18 and keratin 19 [112].

Additional studies on the effects of quercetin on cancer stem cells (CSCs) have shown that this flavonoid represses cell viability and clone formation of CD44-/CD24- BC cells, while reducing metastasis formation in nude mice, which were effects mediated by decreased levels of cyclinD1 and Bcl-2 and by the downregulation of the phosphatidylinositol-3-kinase (PI3K)/Akt/mTOR pathway [113].

The antineoplastic effects of quercetin (corresponding dose of 1311 mg/g per 16 weeks for a man of 60 Kg of BW [114]) have also been described in another mouse model of BC, i.e., in C3(1)/SV40 Tag mice, which displayed a quercetin-induced reduction in tumor volume, as well as revealing a decreased expression of transmembrane protease serine 4, a known marker of poor prognosis for BC [114,115].

At metabolic level, quercetin has demonstrated, in cell and animal models, insulin-sensitizing, antioxidant, and anti-inflammatory effects [116]. However, a clear antiobesity effect has not been observed with quercetin supplementation [117].

#### 4.1.3. Fisetin

Fisetin belongs to the category of flavonols. It is found in various plant products, including apples (26 µg/g), persimmons (10.5 µg/g), grapes (3.9 µg/g), strawberries (160 µg/g), onions (4.8 µg/g), and cucumbers (0.1 µg/g) [118].

Fisetin is known for its senolytic effects [85]. It has been demonstrated that fisetin delays the growth of various cancer cells. Its targets of action are the signaling pathways that regulate tumor cell survival, apoptosis, metastatic, and angiogenic processes. Scientific evidence supports the hypothesis that fisetin could potentially act in cancer treatment and prevention through its epigenetic effects [118]. Some proapoptotic effects are also linked to actions on the cellular membranes [119]. Moreover, fisetin can sensibilize cancer cells to radiotherapy interfering with protection of DNA [120]. In more detail, fisetin (10, 25, and 50 μM) has been shown to induce the apoptosis of breast MDA-MB-453 cancer cells through proteasomal degradation and modulation of the PI3K/Akt pathway [121]. The inhibition of metastasis by fisetin (5 µM) has been suggested through the reduced expression of MMP-2 and MMP-9 enzymes, as observed in BC cell cultures (4T1 and JC cells) treated with this flavonol [122]. In experiments with different models of BC cells, fisetin has shown to be able to inhibit the growth of MCF-7, MDA-MB-468, and MDA-MB-231 triple-negative BC cells, as well as SK-BR-3 human cells overexpressing epidermal growth factor receptor 2 [123]. At the molecular level, fisetin could exert these effects also through apoptosis induction by the inhibition of Aurora B kinase activity. Of note, fisetin has been shown to enhance the cytotoxic effects of cisplatin, 4-hydroxycyclophosphamide, or 5-FU on MDA-MB-468 cells [123]. In accordance with cell-based experiments, in vivo experiments have confirmed that fisetin (20, 40, and 80 μM) is able to counteract tumor growth in a 4T1 orthotopic mammary tumor model by reducing the activation of PI3K/Akt/mTOR signaling pathway [124].

Concerning its role against obesity, fisetin has shown antiadipogenic effects by inhibiting mTORC1 signaling in preadipocytes, and combined with other antiadipogenic phytonutrients, it has proven to be effective in reducing total and visceral fat, lipid levels, and inflammation in murine models [125].

#### 4.1.4. Anthocyanins

Anthocyanins are among the most important groups of purple pigments found in vegetables, but also in fruits. High amounts of anthocyanins, ranging from 94–120 mg/100 g to 442 mg/100 g, can be found in eggplants, black grapes, pomegranates, beets, cherries, mallows, apples, cruciferous vegetables, citrus fruits, berries, and black rice. The more intense their color (reddish or bluish), the higher their anthocyanin content [126]. Growing evidence, although it should be improved, indicates that the major mechanism of their antitumoral effects involves the inhibition of cellular growth and metastasis, including the block of the VEGF signal and pathway, and the degradation of the extracellular matrix. Anthocyanins (500 µg/mL in murine cells) may reverse the multidrug resistance of cancer cells, improving their chemotherapy sensitivity [127].

In particular, black rice anthocyanins have been shown to repress migration and invasion of MDA-MB-453 cells expressing HER2 by downregulating the RAS/RAF/MAPK signaling pathway [128]. In the same BC model, treatment with anthocyanins of blue corn resulted in reduced cell viability and induced apoptosis of BC cells. Of note, the induction of apoptosis was further stimulated by the cotreatment of anthocyanins with the antitumor drug nocodazole [129]. In MCF-7 BC cells displaying high resistance to cisplatin, anthocyanins isolated from *Vitis coignetiae Pulliat* stimulate the sensitivity to this antitumor drug by reducing Akt and NF-kB activation [130]. Antineoplastic effects of black rice anthocyanins (150 mg/Kg/day for three weeks) were observed in vivo, in a murine xenograft model bearing BC cells MDA-MB-453 expressing HER2, whose high levels represent a metastasis risk factor [131]. This study suggested that black rice anthocyanins could repress tumor growth by inhibiting tumor cell motility and invasion, which are two key steps in metastasis [131]. In addition, another in vivo study has shown that treatment with the anthocyanin cyanidin-3-glucoside (6 mg/mL twice a week for 25 days), in a xenograft model bearing BT-474 BC cells (expressing HER2), was able to potentiate the antineoplastic effects of trastuzumab [132].

Regarding metabolic effects, several clinical studies have observed improvements in dyslipidemia or glucometabolic status, as well as a reduction in body weight following supplementation with anthocyanins. Although there is still limited evidence, particularly in humans, the regular dietary consumption of foods containing anthocyanins could be encouraged [133].

#### 4.1.5. Resveratrol

Resveratrol (RE) belongs to the polyphenols and is one of the phytoalexins naturally produced by many plants. Indeed, it is found in grapes, wine, some berries, and oilseeds, tea, and peas. The highest amount is found in grapes (50–100 g/kg net weight), in which it is present only in the skin, while its presence in wine (approximately 0.27 mg/100 g) depends on the cultivar, the geographical origin of cultivation, and the fermentation period.

Besides the defensive function against pathogens such as fungi and bacteria, this non-flavonoid polyphenol is able to exert a powerful antioxidant, antiglycation, anti-inflammatory, neuroprotective and antiaging action [134]. In 4T1cells, RE has been shown to block the cell cycle and induce apoptosis [135]. In experiments with MDA-MB-231 human BC cells, this polyphenol was found to inhibit cell migration and the secretion of matrix metalloproteinase MMP-2 and MMP-9. Importantly, RE was able to reverse the epithelial–mesenchymal transition induced by TGF-β1, which represents a key step for tumor migration [136]. In a murine model bearing MDA-MB-231 human BC cells, treatment with RE reduced the formation of lung metastatic nodules, confirming the antineoplastic properties displayed in in vitro experiments [136]. In MCF-7 BC cells, RE was shown to increase chemosensitivity to the chemotherapy drug Adriamycin, promoting apoptosis by reducing the expression of cyclin-dependent kinases and Bcl-2. Interestingly, modulation of these proteins was suggested to be mediated by resveratrol-induced upregulation of miR-122-5p [137]. In addition, another study observed that RE increases the chemosensitivity of BC cells to the poly (ADP-ribose) polymerase inhibitor talazoparib, leading to reduced proliferation in vitro, as well as decreased tumor growth in SCID-mice [138]. Several studies have shown that RE modulates several genes involved in cancer through DNA methylation. In MDA-MB-231 BC cells, DNA methylation changes were associated with the modulation of transcript levels of tumor-related genes [139]. Other preclinical studies have observed that RE is able to reduce the time of tumor formation and its incidence by reducing angiogenesis and revealing agonist and antagonist actions on estrogen receptors [140].

It is estimated that it is not possible achieve a therapeutic amount of RE with foods [141]. Actually, there is a paucity of data from human trials; however, RE (1 mg/day for 12 weeks) can act on tumor growth by modulating sex hormone metabolism and blocking cell proliferation [142].

It has been observed that RE is able to strongly stimulate mitochondrial activity and oxidative type-I muscle fibers, and regulate the energy balance [143]. However, the benefits observed in preclinical studies have been obtained with much higher doses of RE than those achieved with the usual diet [144]. Moreover, with respect to the antiobesity effects of low-to-moderate wine consumption, human studies have revealed conflicting results [145]. Furthermore, wine contains alcohol, and it has been reported that even very low alcohol consumption is associated with an increased risk of BC recurrence, while the association with a second primary BC or BC-specific mortality is less clear [146,147], and overall, data from the existing studies show a marked heterogeneity.

#### 4.1.6. Curcuma Longa

Although turmeric is not a typical component of the MD, it has been included in this review because it is a spice widely used and studied. Curcuminoids are among the main constituents of turmeric. Curcumin is a polyphenol with anti-inflammatory, antilipidemic, and antioxidant actions. It is able to implement epigenetic modulation and regulation mechanisms by which it could affect various pathologies [148]. Curcumin, even at a dosage of 6 g/day for seven consecutive days every three weeks, is claimed to have anticancer and chemopreventive activity against BC, including antiproliferative effects, antimetastatic effects, the stimulation of apoptosis, blockage of the start of the cell cycle, and the enhancement of several chemotherapeutic effects. These effects may be achieved through various specific mechanisms of action, as well as a complex network of molecular signals that involve the mechanisms underlying cell proliferation. The mechanisms of action include the inhibition of the proliferative and metastatic effects of growth hormone, and the beta-catenin pathway, by triggering onco-suppressor miRNAs and deactivating the oncogenic ones [149]. Unfortunately, curcumin is not highly bioavailable because of its low intestinal absorption, rapid metabolism, and low water solubility. The association of some chemical groups in the molecule can increase bioavailability and efficacy [150]. This may occur with the addition of piperine, which implements liposomal delivery systems, phospholipid complexes, or emulsions with polysorbate [151]. The complexation with proteins and other natural compounds such as quercetin and resveratrol may also improve bioavailability [152].

Regarding the metabolic effects of curcumin, some studies on cell or animal models have shown a beneficial effect on body weight gain, insulin resistance, and lipogenesis [153]. Moreover, some human clinical studies have observed significant weight loss in overweight or obese subjects after supplementation with curcumin [154]. However, further studies are needed to investigate the effects of the dietary use of this compound.

#### 4.1.7. Epigallocatechin-3-Gallate

EGCG is a catechin (flavonol) typical of green tea (27.16 mg/100 mL). It is a polyphenol with anti-inflammatory and antioxidant activity, which is also found in foods such as plums, apples, raspberries, blackberries, walnuts, pistachios, peaches, and avocados. It exerts an anticancer effect by inhibiting some key enzymes involved in the glycolytic pathway and by suppressing glucose metabolism. Some models show EGCG can act as adjuvant of therapies and boost their action against cancer cells. EGCG is a potent antiangiogenic, antioxidant, and anticancer agent, through a modulating action on tumor cells. Preclinical evidence shows that EGCG activity modulates differentiation, proliferation, apoptosis, angiogenesis (40 mg/L of green extract tea), inflammation (2 mg/Kg), and metastasis spreading [155].

In addition, EGCG exerts antilipogenic effects by inhibiting lipid biosynthesis and by stimulating lipolysis [100]. Furthermore, both observational studies and clinical trials have reported a significant association between the regular consumption of green tea or supplementation with green tea catechin and weight loss [145].

### 4.2. Carotenoids

Retinoids are liposoluble compounds related to vitamin A that are present in red and yellow vegetable foods (from 22 to 4000 µg/100 g), eggs, meat, offal, milk, and some fish and shellfish. They are characterized by an unsaturated isoprenoid chain structure. Vegetables are typically a source of carotenoids, some of which are precursors of vitamin A, while animal foods contain bioactive retinoids, synthesized from carotenoids [156,157]. Their bioavailability varies depending on cooking and processing methods and on the presence of fiber and lipids. They are important for visual functions, defense against infections, the maintenance of epithelial cells, reproductive functions, and as an antioxidant, but only if not overdosed. In this case, they appear to have the opposite effect. The daily dietary requirement (indicated as RAEs, that is, retinol activity equivalents) is 12 µg for β-carotene and 24 µg for α-carotene [158]. Carotenoids improve communication at the level of intercellular junctions, which are involved in mechanisms that regulate tumor growth, playing a role in the regulation of cell growth, apoptosis, and differentiation. In fact, gap junction communication is deficient in many forms of cancer, and its restoration leads to a reduction in cell multiplication. They also have anti-inflammatory and immunomodulatory effects. However, further studies are necessary to confirm and define these effects on cancer [159].

A recent study has shown that carotenoids extracted from *Bixa orellana* L. exert antiproliferative effects on MCF-7 cells [160]. Several preclinical studies have observed that the carotenoid fucoxanthin, found in seaweeds and diatoms, acts as a very efficient inhibitor of tumor growth, by modulating signaling pathways involving NF-κB, MAPK, MMP2/9 [161]. Interestingly, in vitro experiments have shown that lutein is able to induce apoptosis in SV40 transformed, but not in normal mammary cells [162]. It also displays antiproliferative effects on MCF-7 and MDA-MB-468 cells [163], thus suggesting promising chemotherapeutic applications. In BC BT474 and SKBR3 cells, cotreatment with all-trans retinoic acid (ATRA) and the antitumor agents tamoxifen or trastuzumab led to apoptosis. Interestingly, in the BT474 cell line, the single treatment with only retinoids such as ATRA, 13-cis-retinoic acid, or N-(4-hydroxyphenyl) retinamide was able to induce apoptosis without the requirement of antiproliferative agents [164]. In another study, cotreatment of the PPARγ ligand rosiglitazone and 9-cis-retinoic acid exerted antiproliferative effects in several BC cell lines such as MCF-7, MCF-7TR1, and SKBR-3. Of note, such effects were not observed in non-cancer breast epithelial cells, indicating that such a multidrug strategy may be a successful approach to selectively inhibit BC cell proliferation [165]. On the other hand, treatment with 9-cis retinoic acid has been shown to stimulate the expression of angiogenesis-related genes, including vascular endothelial cadherin, suggesting potential proangiogenic detrimental effects on tumor growth [166]. Interestingly, retinoids have also been proposed to exert antineoplastic effects by stimulating hepatic detoxification pathways of Bisphenol A, a plastic toxic compound detectable in the food chain, which can elicit BC in various ways, notably activating estrogen receptors and promoting cancer cell proliferation [167,168].

Several clinical studies report low plasmatic concentrations of carotenoids in obesity or related metabolic alterations, while animal studies have found beneficial effects of carotenoid consumption against obesity, insulin resistance, hepatic steatosis, and chronic inflammation. Although these compounds appear to target the adipose tissue, regulating the energy homeostasis indirectly, some studies have hypothesized a direct effect at the brain level [169].

#### Lycopene

Lycopene is a natural antioxidant of the carotenoid family, found in tomatoes, watermelon, pink grapefruit, apricots, grapes, papaya, carrots, and pumpkins. The highest concentrations are found in ripe tomatoes, and their bioavailability and adsorption can increase when ripe tomatoes are cooked or heated. Among carotenoids, lycopene is the most powerful and exerts an anticancer effect, particularly in ER-negative BC. This action is exerted through a cytotoxic and antiestrogenic effect. Moreover, it reduces cell proliferation and blocks the tumor cell cycle. These effects are not observed in healthy subjects [70]. Lycopene is a carotenoid with two unconjugated double bonds that provide a powerful antioxidant property, even at near-physiological levels (0.52 µM), by inhibiting cell proliferation, inducing apoptosis, and suppressing metastasis, but also by potentiating the effect of anticancer drugs (paclitaxel, docetaxel, cisplatin, and adriamycin). The treatment that tomatoes undergo for food use (crushing and/or heat treatment) results in cis-isomerization, which increases their bioavailability compared to the fresh product; an enhancing effect is also due to the synergy between lycopene and other dietary nutrients (lipids, proteins, and polysaccharides, quercetin, curcumin, and genistein) [170]. The lycopene content in ripe tomatoes ranges from 0.88 to 7.74 mg/100 g, while the plasma concentration (after a daily intake of 10 mg of lycopene) is around 0.52/0.6 µM. Lycopene inhibits free radicals by reducing oxidative damage and inflammatory response; counteracting lipid peroxidation; protecting lipids, RNA, and DNA from oxidation by preventing mutations; and inhibiting the growth of cancer cells and unhealthy cells [170,171]. High levels of lycopene as well as serum α-carotene, β-carotene, and lutein/zeaxanthin were associated with a lower risk of BC among all ER- or progesterone receptor-positive subtypes [172].

An inverse association has been shown between lycopene intake and BC risk, which seems to be greater in lean women than in overweight and obese women. This finding is probably due to the fact that lycopene is fat-soluble and easily stored in adipose tissue, but less available at the plasma level. In that study, the subjects showed an association between the dietary intake of carotenoids and reduced risk of BC, even for the aggressive and lethal forms [173]. Sisti et al. demonstrated the inverse association only in postmenopausal women, and attributed this result to the interaction with hormonal factors [174]. It is known that a high BMI in postmenopausal women increases BC risk. Llanos et al. observed that the consumption of lycopene in postmenopausal obese women determined an increase in circulating levels of adiponectin, which regulates glucose homeostasis and fatty acid metabolism. Thus, it can be hypothesized that lycopene can improve insulin sensitivity and the lipid profile, with potential protective effects on BC [175].

Fukuschi et al. observed that a six-month administration of tomato juice at a dose of 160 g/day produced an early increase, during the first month, of serum lycopene content, which reached its highest values at 3 and 6 months and returned to baseline levels at 12 months. This long-term intake of tomato juice contributed to the recovery of early and late skin adverse events caused by radiotherapy that women underwent after breast-conserving surgery [176].

Although the main effect of lycopene, in preventing carcinogenesis, can be attributed to its high antioxidant property, through which it can suppress the tumoral promotion and progression, other mechanisms have been proposed. Among these, there are the regulation of growth factor signaling, the induction of apoptosis, cell cycle arrest, the reduction of metastasis, angiogenesis, and cell invasion. Lycopene treatment has been shown to reduce proliferation and increase the rate of apoptosis of MCF-7 human BC cells by upregulating levels of p53 and Bax transcript [177]. In accordance with these data, in MDA-MB-468 cells, lycopene administration results in the reduced activation of Akt and mTOR, involved in cell proliferation, and increased levels of the proapoptotic protein Bax [178]. In addition, lycopene may exert antitumor effects by down-regulating the NF-κB signaling pathway [179]. These effects are certainly demonstrated by preclinical evidence but need further confirmation in the clinical field, which, however, appears to be promising [178].

To summarize, lycopene exerts antioxidant, anti-inflammatory, and anticancer effects. In addition, it appears to have a protective role against obesity and diabetes, although prospective studies are needed to establish the possible therapeutic effect against metabolic disorders [180].

### 4.3. Folates

Folate is abundant in some vegetables and fruits, in particular leafy vegetables. In the body, it has the function of donating a one-carbon group in DNA methylation and synthesis [181]. An intake of 100 micrograms per day (found in about 160 g of spinach, including cooked spinach) has been associated with a 10 percent decrease in BC risk among women who drink moderate amounts of alcohol (a risk factor for breast cancer). [182]. However, although some studies suggest a reduction in cancer risk [183,184], other studies reported an increase. The influence of folate intake on BC risk may also be linked to menopausal status. Indeed, the study by Ren et al. showed a linear negative correlation between folate intake and reduced breast cancer risk, that is, each 100 μg/day increase in folate reduces BC risk by 2%, only in premenopausal women [185]. Moreover, Frederick et al. found increased folate concentration in the breast tissue of premenopausal women, while in the same study, obesity was associated with low serum folate levels. The authors believe that this finding may explain the discrepancies reported in the literature as regards the effects of folate on breast cancer. In fact, these authors hypothesize that folate may have a preventive effect in premenopausal women through an epigenetic mechanism, and a carcinogenesis-supporting effect in postmenopausal women by promoting tumor growth through DNA synthesis [186].

Epigenetic changes (such as DNA hypomethylation and hypermethylation) have been observed in some cancers, including BC, and several studies have shown an overexpression of folate receptors (FR) in different tumor surface types, including BC. This overexpression tends to increase as the disease progresses, which is considered a negative prognostic factor. Therefore, such receptors have been proposed as a useful target in personalized therapy against BC [187,188]. This target is particularly useful in triple-negative BC (TNBC), in which reduced folate intake or the use of antifolate drugs result in mitochondrial dysfunction, which reduces metabolic plasticity [189].

A very strong inverse association between folate intake and survival was also demonstrated in the prospective study by Harris et al. among patients with ER-negative BC. This effect could be attributed to the fact that insufficient folate intake can result in an improper methylation of DNA and RNA, thereby altering their integrity and repair, which are critical factors in cancer survival [190].

Moreover, a U-shaped relationship has been observed, indicating that increased folate intake before the development of preneoplastic lesions may prevent tumor formation, but the supplementation with folate after the lesions are present may promote tumor progression [185].

Furthermore, folate supplementation seems to be able to reduce chemotherapy drug toxicity, but an excessive supplementation increases drug resistance [191]. Such a window would need to be better defined, but it is likely that an adequate folate intake could fall within the ideal dose window to better balance the two effects.

Notably, McEligot observed that a plasma folate concentration of 25.2 nmol reduces the risk of death after BC diagnosis by about 30 percent, while 40.6 nmol of folate reduces the risk of death after BC diagnosis by more than 50 percent. However, a dose of 1000 μg appears to lose this effect, and it may also increase the risk [192].

### 4.4. Indole-3-Carbinolo (I3C), Di-Indoylmethane (DIM) E Sulforaphane

DIM and its precursor, I3C, are active compounds present in Brassicaceae or vegetables derived from them: broccoli, cabbages, Brussels sprouts, and savoy cabbage. Anticancer properties associated with cruciferous vegetables are mainly due to glucosinolates, a family of secondary metabolites, which are typical only of this family and play a dominant role in defending plants against parasites. Glucosinolates are beneficial for general health due to their ability to arrest the initial stages of carcinogenesis. The effects have been shown by the suppression of the proliferation of several cancerogenic cell lines; the induction of apoptosis, but not in healthy cells; and the modulation of estrogen metabolism, thereby reducing cancer risk in hormone-sensitive cells [193].

It is unlikely that the quantities present in the diet are sufficient to have therapeutic effects; thus, a supplementation can be considered [194]. Supplementation with red cabbage extracts containing isothiocyanates has shown interesting results in modulating gut microbiota. In particular, an increased production of butyric acid has been observed, and due to the possible regulation of food intake mediated by short-chain fatty acids, this finding could pave the way for a promising treatment for overweight or obesity [195].

### 4.5. Capsaicin

Capsaicin is a chemical compound found in varying concentrations in plants of the genus *Capsicum* such as hot pepper, which gives them their characteristic irritant power. Chili peppers are the main source of capsaicinoids in nature. Capsaicin content has been found to range from 2.19 to 19.73 mg/g dry weight of Capsicum fruits [196]. It could be used clinically to relieve pain, prevent cancer, and promote weight loss due to its antioxidant and anti-inflammatory properties. Capsaicin manifests anticancer activity through delicate epigenetic molecular mechanisms. On one hand, it inhibits the expression of vascular-endothelial growth factor) (VEGF), which is responsible for tumor vascularization and the angiogenic process that fuels the cell growth and the development of metastasis [197]. On the other hand, it facilitates apoptosis and induces antiproliferative effects through the reduction of FBI-1 protein expression. These effects were observed after intraperitoneal administration of 10 mg/Kg every three days for a total of 21 days [198].

### 4.6. Dietary Fiber and Whole Grains

Consumption of whole grains has been shown to be able to reduce the risk of obesity, diabetes, and cardiovascular diseases, as well as prevent cancer. Whole grains could influence obesity by promoting the sense of satiety, reducing calorie intake, and managing body fatness due to its high content of fiber, which can reduce energy density [199].

The high fiber content in whole grains could significantly control insulin resistance and the expression of insulin-like growth factors [200], and enhance metabolic flexibility [201]. In addition, decreasing body adiposity and insulin resistance may contribute to reducing cancer risk.

Whole grains, as well as fruit and vegetables, are rich in phytochemical substances such as phenolic acids [202], carotenoids, alkylresorcinols, phytosterols, lignans, anthocyanins, vitamin E, and polysaccharides [203].

Several experimental studies have shown that the bioactive compounds of whole grains exert anticancer activity through the inhibition of cellular multiplication, the modulation of the immune system, and the inhibition of tumor metastasis. Whole foods are sources of isoflavones like phytoestrogens, which may influence hormone levels and activity. Phytoestrogens combined with dietary fiber may decrease internal estrogen concentration, inhibit cancer development, and weaken the early expression of markers of BC risk, although possible negative effects should be carefully evaluated [204].

As for fiber intake, a protective association between fiber intake and BC risk is around a 12% reduction. In addition, an increase of every 10 g/d of dietary fiber is associated with a 4% risk reduction [205,206].

However, regarding the association between whole grain consumption and BC risk, discordant and inconsistent results are reported, possibly due to the different study design and the lack of adjustment for risk factors. Moreover, epidemiological studies have considered “whole grains” as a general parameter of consumption without specifying the different grain species and growing regions, and without accounting for the varying concentrations of bioactive phytochemicals that are associated with these factors.

There are not many studies investigating the effect of fiber intake in the postdiagnosis period of BC. In a randomized controlled trial, BC survivors who increased vegetable consumption by 65%, fruit consumption by 25%, and fiber consumption by 30% did not have reduced mortality and recurrences after a follow-up of 7.3 years [30]. Another study, despite the weak and not significant inverse association between fiber intake and BC mortality and recurrence, suggested that the positive trend should be considered because fiber intake reduces estrogen levels, modulates insulin resistance, and reduces levels of inflammatory cytokines such as IL-6, TNF-α-R2, and C-reactive protein [207]. The study by Villasenor et al. supports this hypothesis, as patients who had consumed more than 15 g/day of insoluble fiber, 24 months after diagnosis, had a 49% reduction in the likelihood of having elevated PCR concentrations compared to patients who consumed amounts of less than 5 g/day [208]. More recently, Zengul et al. did not find any correlation between dietary fiber and estrone or estradiol levels, but they found an inverse association between soluble fiber and estradiol levels. High fiber consumption was correlated with low levels of *Clostridium hathewayi* sp., *Clostridium* (*Erysipelotrichaceae* family), and increased *Bacteroides uniformis*. All of these bacteria are associated with increased ß-glucuronidase activity, which is important for luminal hormone metabolism [209].

In conclusion, there are no conclusive data regarding the beneficial effects of fiber intake on the prognosis of BC; however, the observed positive trend would merit further investigation.

### 4.7. N-3 Fatty Acids (or Omega-3)

The most abundant n-3 PUFA in the diet is alpha-linolenic acid (ALA), usually found in plants, while the n-3 PUFAs eicosapentaenoic acid (EPA) and docosahexaenoic acid (DHA) are widely found in cold-water fatty fish such as salmon (1.04 g/100 g), sardines (992 mg/100 g), and mackerel (3 g/100 g). EPA and DHA can also be synthesized within the body from ALA through the activity of desaturase and elongase, which have a higher affinity for ALA than n-6 linoleic acid (LA), typical of some plant oils. However, modern dietary habits lead to a much higher intake of LA, with a consequent prevalence of n-6 arachidonic acid synthesis compared with EPA and DHA. The synthesis of long-chain PUFA depends on the activity of elongases and desaturases, which shows considerable variations among individuals [210,211,212,213].

The suggested intake of EPA+DHA is 500 mg/day [84] or from 0.5 g at 6 months old to 1.1/1.6 g for an adult person [209].

Large amounts of LA are found in vegetable oils, seeds, and nuts, while ALA is found mainly in leafy vegetables, chia seeds, nuts, seed oils, flaxseeds, and soybean oil, with varying amounts (from 0.5 g/100 g in soybeans oil to 10 g/100 g in nuts, and 50 g/100 g in flaxseeds). As for n-3 PUFAs, they can be conjugated with various amines, including dopamine and serotonin, and be further metabolized through enzymatic and nonenzymatic pathways [214]. Enzymatic mechanisms, through the action of cyclooxygenases (COX), lipoxygenases, and cytochrome P450, lead to the formation of oxygenated derivative compounds. All these products have been shown to be effective in inhibiting tumor growth, both in vivo and in vitro. Additionally, they exhibit immunomodulatory effects by attacking tumor cells while preserving healthy cells [214]. The peculiarity of unsaturated fatty acids such as n-3 PUFAs is that they are peroxidable and easily incorporated into membrane phospholipids, especially in BC cells with compromised membrane integrity, by breaking down or sequestering membrane proteins. Furthermore, they neutralize reactive oxygen species (ROS) produced by cancer cells [215]. Regarding the other compounds, n-3 PUFAs can modulate the expression of genes involved in cell death and lipid metabolism, while EPA and DHA also manage to produce metabolites that reduce inflammation. In particular, EPA, due to the action of COX and LOX, gives rise to prostaglandins, thromboxanes, and leukotrienes with anticancer and anti-inflammatory properties, thus inhibiting the growth and invasion of tumor cells [216]. The enrichment of the membranes with EPA and DHA induces an increase in resolvins and protectins, which are important for healthy cells by protecting them from toxic chemicals such as chemotherapy drugs [215,216,217]. The only supplement recommended by ESPEN for cancer cachexia is n-3 PUFAs [218], which can be of particular interest in TNBC [219]. Interestingly, omega-3 free fatty acids have been suggested to modulate the insulin signaling pathway and affect BC cell growth. Increased levels of insulin represent a risk factor for BC, and in vitro studies have shown that omega-3 free fatty acids are able to dampen the activating phosphorylation of protein AKT, which is a downstream target of the insulin receptor, and to inhibit the proliferation of MCF-7 cells [220]. DHA and EPA, by acting as ligands of PPARγ, are able to modulate PPARγ transcriptional activity with subsequent gene expression modulation, which inhibits mTOR signaling and stimulates autophagy [221]. Of note, DHA-dopamine and EPA-dopamine conjugates have been shown to reduce the in vitro viability of BC cell lines such as MCF-7 and MDA-MB-231, with no effects on breast non-cancerous epithelial cells. Both compounds have been found to stimulate the transcription of the autophagy regulator Beclin-1 by PPARγ, with potential antineoplastic effects [222].

## 5. Food Synergy

The dietary supplement (DS) industry was developed on the idea that individual nutrients, even when administered in isolated form, have the same effects and health benefits as when they are consumed as a part of whole foods. Nonetheless, some studies have shown that many DSs do not work as intended or may have adverse effects. Moreover, the aspect of synergy can be lost because nutrients can mutually influence their absorption and functions [223].

Food synergy refers to the concept that the constituents of foods can have additive or even more than additive effects when consumed within their natural food matrix, or as part of a meal that reflects more complete dietary patterns. It is referred to as a “complicated system acting in concert on health”. Unfortunately, synergies are not easy to demonstrate experimentally. To better understand them, it would be useful to carry out a series of large studies, over both short and long periods, on different geographic areas, ethnicities, ages, and genders, taking into account various dietary and food patterns [224].

The binding and other native molecular modifications of food are very relevant to understanding nutrition. If molecules require coupling for their functionality, eating them simultaneously could increase the probability of this coupling. Eating different food within a 24 h period may be sufficient for coupling to occur in the digestive tract or at the systemic level. In this way, the alimentary synergistic theory supports the consumption of a variety of foods rich in micronutrients [224].

The MD, which comprises a variety of foods predominantly of plant origin, exerts its beneficial effects on health due to the synergistic activity among the various nutrients rather than to individual components [225]. In addition, the positive association of the MD with healthy aging compared with other diets could be attributed to its integrated food-based approach, emphasizing the synergy among various foods, rather than a nutrient-based one [226].

Phytochemical extracts of fruits and vegetables have demonstrated strong antiproliferative and antioxidant effects. These effects are mostly attributed to the synergistic activity of the various phytochemicals present. This underscores the idea that no single molecule can replace the natural combination of phytochemical substances in fruits and vegetables to obtain health benefits. Furthermore, antioxidant or bioactive compounds are best obtained through the consumption of whole foods, rather than from costly dietary supplements [227].

Different species and varieties of fruits, vegetables, and grains present various phytochemical profiles. The combination of blueberry, orange, grape, and apple has been shown to have an antioxidant synergistic effect. The dose–response curve of antioxidant activity demonstrates an increased effect after combining of the four fruits [228].

Micronutrient blending enhances metabolic targets by amplifying the biological impact with lower component doses, due to nutrient synergy. Synergistic blending of micronutrients could effectively target multiple aspects in cancer development and progression, even when metastasis has occurred. Therefore, it could be a promising and cost-effective approach to consider in cancer therapy [229].

Parizad et al. reported that phenolic compounds and anthocyanins in colored grains are present in sufficiently high doses to be considered as good bioactive compounds. Although to different degrees and with different specificity, all extracts considered in this study showed anti-inflammatory and inhibitory effects on pancreatic α-amylase and intestinal α-glucosidase. The beneficial effects of most of these extracts appeared to be higher than those provided by bioactive-equivalent concentrations of extracts from other sources. These findings suggest that both enzymatic inhibition and anti-inflammatory effects could be attributed to a synergy between components rather than a particular compound [230].

Interestingly, it has been observed that a diet rich in polyphenols can promote detoxification processes and limit the intake of advanced glycation end products (AGEs), which are inflammatory substances formed during high-temperature cooking. Conversely, raw products, and especially fruit and vegetables, which are abundant in the MD, contain low AGE concentrations. AGEs can play a role in the initiation and progression of cancer, and restricting their intake may help prevent and support cancer therapy [231].

Finally, an important element to consider is dietary diversification. It derives significant benefits from synergistic interactions that improve micronutrient bioavailability. However, dietary strategies should consider varying eating habits to ensure better acceptance by the consumers [232].

### Pomegranate

Besides the food synergy exerted by the consumption of various nutrients in the same meal, there are some functional foods that manifest a synergistic effect per se. An example of these foods is pomegranate, which has been studied for its potential beneficial effects on various medical conditions such as male infertility, immunological and neurological dysfunctions, microbial infections, ulcer, inflammation, diabetes, arthritis, obesity, cardiovascular disease, and cancer [233].

As regards BC, pomegranate has been found to exert chemopreventive effects on experimentally induced mammary tumorigenesis through the suppression of cell proliferation and the induction of apoptosis [233].

The aforementioned properties have been attributed to the presence of a large number of phytochemicals such as flavonols (quercetin), conjugated fatty acids (punicic acid), hydrolyzable tannins and related compounds (gallic, gallacic and ellagic acids, ellagitanin, punicalagin, pedunculagin), flavonoids (anthocyanins and catechins), and flavones (epigenin and luteolin) [233].

The seeds and arils account for 50% of the fruit. The seeds have a high content of anthocyanins (pelargonidin-3-glucoside, cyanidin-3-5-diglucoside, pelargonidin-3,5-diglucoside, delphinidin-3,5-diglucoside, delphinidin-3-glucoside and cyanidin-3-glucoside), and hydrolyzable tannins (esters of glucose, punicalagin, gallacic and ellagic acids, punicalin and pedunculagin). On the seed coat are present malic, citric, and ascorbic acids, while arils are rich in flavonoids and phenolics. The antioxidant activity of pomegranate anthocyanins is much higher than that found in green tea and red wine [234].

Moreover, pomegranate is rich in various compounds, such as flavonoids, ellagitannins, ellagic acid, 3-glucosides/3,5 delphinin, cyanidin and pelargonidin, which have been recognized for their anti-inflammatory, antioxidant, and anticarcinogenic effects. Pomegranate reduces tumor volume by reducing cell growth. Pomegranate extracts (0.8 mg/kg/day) reduce mRNA levels for Sp1, Sp3, and Sp4 implicated in angiogenesis, inflammation and cell proliferation and Sp-regulated genes such as VEGF, survivin and NF-kB [235].

In particular, it has been reported that pomegranate extract has the ability to inhibit the expression and activity of NF-kB and to reduce the expression of genes involved in cell migration and invasion in TNBC. It also reduces the serum concentration of VCAM-1, but not in TNBC cells. The metastasis-inhibiting effects could be due to the reduced expression of fibronectin, which drives cell migration along particular pathways [236].

Finally, it also reduces VEGF expression, counteracting the angiogenesis, which is characteristic of tumor processes [237].

Polyphenols are present in pomegranate in much higher amounts (approximately 3.8 mg/mL gallic acid equivalents) than in other fruits such as black cherry, grape juice, and orange, which have ranges from 0.46 to 2.6 mg/mL gallic acid equivalents [238]. Polyphenols in fermented juice and pomegranate seed oil succeed in inhibiting the enzyme implicated in the conversion of estrone to estradiol (17-bhydroxysteroid dehydrogenase type 1) expressed in many BC cells. Moreover, these compounds have been found to inhibit cyclooxygenases 2 (COX2), also expressed in BC cells. It contributes to tumor development by stimulating prostaglandin synthesis, which in turn stimulates aromatase. Furthermore, polyphenols have been shown to reduce the expression of two proteins involved in cell motility, namely RhoC and RhoA [239].

However, some studies have shown that the effects of the polyphenols in pomegranate can be attributed to their action through several pathways, including metabolites produced by the gut microflora; thus, its effects may be dependent on it [239].

Copper is an important metal ion involved in the redox activity within cells. An increase in copper levels has been observed in the tissues and serum of individuals with malignant tumors. Anthocyanins, found in pomegranate, can cause copper-mediated cell death by increasing its ability to generate ROS near DNA [240].

The phytochemicals in pomegranate are absorbed in the gastrointestinal tract. The metabolites produced by the intestinal flora appear to be primarily urolithins. Therefore, variations in the intestinal microflora could be responsible for differing concentrations of these metabolites. Recent evidence suggests that urolithins, especially urolithin A and urolithin B, possess potential beneficial effects against BC. Specifically, antiproliferative and estrogen-modulating activities have been observed [241].

The no-observed-adverse-effect level (NOAEL) of pomegranate extract was set at 600 mg/kg body weight/day. After consuming 180 mL of pomegranate juice, containing 318 mg of punicalagins and 25 mg of ellagic acid, 31.9 mg/mL of ellagic acid was found in plasma within an hour, but vanished after 4 h. Instead, urolithin derivatives remained 48 h postconsumption. No punicalagins were found in the plasma after one hour [242].

Concentrations of 2.35 µM and 4.7 µM urolithin B, or 4.7 µM gallagic acid, are needed to inhibit aromatase [234].

## 6. Nutrigenomics

The term “epigenetic diet o nutrigenomics” was coined to describe the consumption of foods that can influence epigenetic mechanisms, which are non-genomic factors that modulate gene expression without altering the DNA sequence. Such a diet can help protect against the aging process and cancer. The mechanisms involved in the epigenome modulation include changes in DNA methylation, covalent modifications of histones, or RNA modifications, and they play a critical role in tumorigenesis [243,244].

Various foods, such as soy, green tea, cruciferous vegetables, and fruits and vegetables in general, have been demonstrated to have positive effects on the epigenome. These foods can be included in an “epigenetic diet”, a dietary model that can be used therapeutically to maintain health and prevent disease [245].

The MD is particularly rich in foods with nutraceutical effects: bioactive compounds that promote health and longevity, by acting directly or through epigenetic effects, such as DNA methylation [101].

However, it is not always feasible to consume the optimal amount of phytoderivates and nutrients that promote protein synthesis through gene transcription, while also trying to optimize genomic susceptibility by enhancing the positive aspects and discouraging the negative ones. The concept of nutraceuticals was born as a mix of substances capable of interacting with the genome based on the environmental structure (phenotype). Nutraceuticals play the role of cellular and functional modulators, helping to optimize the physiological functions [210]. Recent findings in understanding the mechanism of nutrigenetics, nutrigenomics, and nutraceuticals have led to the concept of superfoods, which are foods that can have a positive impact on genetic expression [101,102,149].

## 7. Intestinal Flora

The gut microbiome, which refers to the genetic make-up of the microbiota, is influenced by many factors, including genetic factors. Instead, diet is the main modulator of the composition and diversity of the gut microbiota, namely the population of microorganisms colonizing the gut, and this interplay could significantly impact multiple health outcomes. Dietary pattern influences gut microbial composition in patients with overweight, obesity, or type 2 diabetes as well as in normal-weight individuals [246,247,248]. More specifically, microbial changes are linked to the intake of both plant- and animal-based foods, as well as to the intake of fats, proteins, carbohydrates, fiber, and phytochemicals [31].

In healthy individuals, the GM is characterized by a high species richness and diversity and its associated with longevity [249]. Specific species support a healthy state by competing against pathogenic agents, helping in the digestive process and producing bioactive components and short-chain fatty acids [250]. The balance present in the intestinal flora of a healthy individual helps to counteract inflammation.

According to the literature, more than 30% of all cancers are related to food intake, including 50% of breast carcinomas. In diet-associated BC, microbiota-mediated mechanisms have been suggested as possible modulators of carcinogenesis and tumor aggressiveness [251]. It has been estimated that 15–20% of cancer, including BC, can be etiologically linked to the overgrowth of bacterial pathobionts or their inflammatory metabolites [252].

Furthermore, recent studies show that the GM of women with BC differs from that of healthy women, and specific bacteria may be linked with different reactions to the therapy [32,253].

Several in vitro and in vivo studies show considerable evidence that diet, probiotics, and prebiotics could exert anticancer effects in BC and could also be utilized as conventional BC treatment adjuvants. Moreover, anticancer treatments reduce the barrier function in mucosae. The GM can protect from mucositis due to chronic inflammation linked to altered permeability [251,254].

In animal models, it has been shown that the consumption of a WD or MD can modulate the microbiota and metabolite profile of mammary glands. Notably, it has been observed that consuming an MD may potentially confer anticancer properties. These findings suggest that changes in metabolites may reflect local fluctuations in microbial metabolism within the mammary gland, which could be influenced by dietary patterns [72].

Moreover, several studies suggest that probiotics play a role in reducing the size and aggressiveness of cancer in a TNBC murine model. Dietary polyphenols act not only as antioxidants, but also as modulators of the GM, and in some cases as antibiotics. Although their structure limits their bioavailability through intestinal uptake, microbial hydrolysis converts dietary polyphenols into more absorbable compounds that can exert anti-inflammatory and anticancer effects [72].

The preventive and potentially therapeutic role of probiotics and prebiotics may be related to alterations in estrogen metabolism; systemic immune regulation, including modulation of immunotherapy; and epigenetics regulation. Bacterial metabolites reduce tumor growth and propensity to metastasis. Moreover, GM modulation can alleviate side effects of chemo- and radiotherapy [255].

Finally, some evidence has shown that symbiotic supplementation can lead to a reduction in inflammation and oxidative stress markers, as well as an increase in antioxidant enzyme activity in patients with breast cancer-related lymphedema [256,257,258].

## 8. Discussion

There is some evidence that indicates that nutrition may influence the clinical outcomes of BC, particularly in terms of recurrence rate and mortality [28]. Although the specific effects of diet are not fully understood, it has been suggested that a healthy lifestyle, including regular physical activity along with a diet rich in plant-based and unprocessed foods and low in saturated fats and meat, may improve survival and quality of life for patients with BC [259].

Moreover, it has been observed that after diagnosis and treatment of BC, patients tend to gain weight, and only a minority of them meet the lifestyle recommendations including physical activity and proper nutrition [260]. It has also been reported that weight gain is associated with an unfavorable medical course of BC and an increased risk of recurrence and mortality [261]. More specifically, for every five-unit increase in BMI prediagnosis, there is a reported 10% increase in overall mortality and a 7% increase in mortality specifically for BC. Furthermore, the presence of abdominal obesity is linked to a 23% increase in overall mortality, regardless of the BMI value [262]. In particular, the higher mortality can be linked to the increase in visceral fat and the subsequent hyperinsulinemia and inflammatory status, both associated with an unhealthy diet [263,264,265]. Given that chronic inflammation and obesity-related hyperinsulinemia have been associated with poor clinical outcomes in BC, an anti-inflammatory and antihyperinsulinemic dietary approach could potentially improve the prognosis of BC [265,266].

Additionally, the long-term intake of healthy foods promotes the well-being of the gut microbiota, which in turn has positive effects throughout the body [267,268]. It has been observed that the GM composition plays a role in the development of different types of tumors, including BC [269]. Moreover, several clinical studies have demonstrated that the status of GM could influence the response to the specific treatment for BC, in terms of efficacy and side effects. Nonetheless, most of these studies have been performed on small patient samples and have not considered the combined effect of nutrition [32].

In terms of nutrition, a systematic review and meta-analysis of various dietary patterns found a significant reduction in mortality among BCS who followed a higher diet quality. A possible explanation of this result could be linked to the anti-inflammatory potential of this diet, as suggested by the low levels of C-reactive protein independent of the BMI values [270].

With regard to high-quality diets, the MD has been shown to effectively reduce comorbidity, recurrence rate, and mortality in BC, representing a valid nutritional model for patients during and after treatment.

In particular, a systematic review and meta-analysis of observational studies and randomized controlled trials has shown that high adherence to the MD is associated with a reduced risk of all-cause mortality among BCS [271].

Furthermore, the EPIC cohort study prospectively evaluated the role of the MD on BC survival. Results show that a high adherence to this diet before diagnosis was significantly associated with a reduction in overall mortality when compared to a low adherence, while no significant difference was found when compared to a medium adherence. This result was stronger among postmenopausal women and in the presence of metastatic tumors [272].

The benefits of the MD seem to be related not only to the balance of macronutrients, but also to the presence of several micronutrients, which, despite being present in small amounts, can contribute to the positive effects of this diet [215]. The presence of micronutrients in fruits, vegetables, whole grains, spices, flavorings, and EVOO exerts important antioxidant, anti-inflammatory, and anticancer effects [82].

With regard to the specific compounds, an abundant class in the MD are phenols, which include quercetin (found in red onions, capers, cruciferous, tomatoes, and asparagus), fisetin (found in apples, persimmons, grapes, strawberries, onions, and cucumbers), anthocyanins (found in vegetables and fruit such as black grapes, cruciferous, apples, pomegranates, and red and blue fruits), and EGCG (found in green tea, plums, blackberries, pistachios, walnuts, and raspberries). Besides their antioxidant and anti-inflammatory effects, phenols can also exert anticancer effects by preventing cell growth, metastasis, and angiogenesis, as well as promoting apoptosis. Other beneficial compounds in the MD are carotenoids, which include retinoids and lycopene. They are found in vegetables and foods characterized by a typical yellow or red color, such as carrots, fish, offal, milk, eggs, papaya, watermelon, and apricots. These compounds have anti-inflammatory, immunomodulatory, and antiangiogenic properties. Moreover, they contribute to epithelial cell maintenance and regulate cell growth, differentiation, and multiplication. Finally, folates, capsaicin, and curcuminoids, found in vegetables, fruits, and roots, have been found to exert some anticancer effects [273,274,275]. Among other compounds, it has been observed that 9 mg of oleocanthal, a phytochemical similar to ibuprofen found in 50 gr (or approximately 55 mL) of EVOO, can exert a mild anti-inflammatory and antioxidant effect, which may protect against thrombosis and platelet aggregation [276].

It is very important to note that nutrients from whole foods can have a greater impact on body functions than when consumed as supplements. This is because within the food matrix, their potential can be enhanced by reciprocal interactions of the various compounds. Indeed, dietary effects of the distinct components may be inadequate to detect, but their additive impact may be quite large. Different foods can interact synergistically to counteract oxidative stress and inflammation, as well as cancer-related mechanisms. These considerations suggest that the development of nutraceutical combinations, even in the form of foods or beverages, could be an effective strategy against chronic degenerative diseases, including cancer [277].

The MD represents a good example of food synergy, being a food-based approach characterized by a balanced combination of macronutrients and micronutrients, which in turn favor the interaction among them. These features are considered the main reason to explain the multiple health benefits associated with this dietary pattern [226].

As regards BC, a high adherence to the MD is associated with benefits in terms of both prevention and improvement of clinical outcomes [271,278]. The combination of foods with nutraceutical properties and the richness in bioactive compounds of many typical foods of the MD such as fresh vegetables, fruit, and EVOO could exhibit multiple anticancer effects. Besides the antioxidant and anti-inflammatory activities, these compounds may also modulate molecular mechanisms associated with cell survival, proliferation, differentiation, migration, and angiogenesis [84].

Another point of interest regards the potential effects of nutrition on the symptoms that commonly occur during cancer treatment or progression.

Several therapeutic options such as endocrine therapy, immunotherapy, chemotherapy, surgery, and radiation therapy are used to cure or prolong the patient’s life, but each therapy can have unique side effects related to its use. In addition to the treatment-related side effects, patients can experience several concomitant physical and psychosocial effects. These effects can include fatigue, nausea, pain, sleep disturbances, depression, anorexia, anxiety, cognitive impairment, and metabolic alterations. A symptom cluster is defined as a group of at least two coexistent symptoms that are correlated and occur during cancer therapy or disease progression. The occurrence of symptom clusters has been found to be associated with various factors, including age, socioeconomic status, the presence of comorbidities, and type of cancer treatment. Some of these symptoms can persist for many years in BC survivors, even after the completion of therapies. These manifestations are a significant source of distress and can affect the daily routine and quality of life [279].

Some researchers have found that BC survivors who follow a better diet quality tend to experience less total fatigue, a common symptom among cancer patients. Consuming foods that are rich in anti-inflammatory nutrients such as n-3 PUFAs, vitamin C, carotenoids, and vitamin A has been associated with less persistent fatigue [280,281,282].

The MD not only offers anti-inflammatory benefits, but it may contribute to reducing cancer-related fatigue, as well as improving the GM composition. Moreover, compared to more restrictive diets, the MD is easier to follow [283].

Significant associations have also been observed between cognitive recovery and dietary patterns following cancer therapy. Specifically, a higher intake of fruits, vegetables, tea, fish, and vitamin B and E supplementation has been linked to higher cognitive scores. Conversely, the consumption of alcoholic beverages has been associated with a reduction in cognitive scores [284].

Several spices, berries and herbs, including commonly consumed ones like onion, garlic, pepper, thyme, and curry, contain many bioactive substances that are known to possess multiple medical properties that can be useful against BC [285].

An educational dietary intervention designed to increase adherence to the MD among BCS reported a significant increase in the consumption of several spices and herbs compared to baseline levels. One of these spices, cinnamon contains various compounds known for their antitumor and anti-inflammatory effects [286]. As for other natural products, ginseng and guarana have been shown to improve cancer-related fatigue, black cumin may have antidepressive effects, while rosemary seems to inhibit cell proliferation. Moreover, ginger can alleviate chemotherapy-induced nausea and vomiting by attenuating the production of chemokines and cytokines involved in gastric inflammation [287].

A randomized study showed that nutritional interventions, when performed during the first three cycles of chemotherapy for women with BC, can lead to positive effects against the occurrence of some adverse events from treatment such as emesis, nausea, and anorexia. Furthermore, it may decrease the occurrence of leukopenia and other gastrointestinal symptoms during treatment. Overall, these benefits contribute to an improved quality of life. These results underscore the importance of nutritional support during chemotherapy. Proper nutrition can also prevent the possible impairment of nutritional status during therapy, potentially minimizing clinical setbacks, including the risk of treatment interruption [288].

Maintaining nutritional status is crucial for BC patients, many of whom show overweight or obesity and need weight loss programs that do not affect lean body mass [289]. Such programs should incorporate long-lasting multimodal lifestyle interventions that include proper nutrition, regular physical activity, and behavioral changes [27].

The MD is strongly recommended as an example of a high-quality diet that promotes a healthy lifestyle. Moreover, a recent review reported an inverse association between high adherence to the MD and overweight, obesity, and various obesity-related chronic diseases [33]. In this regard, it can help counteract the weight gain frequently observed in BC patients, potentially contributing to improving their prognosis [290,291]. Indeed, obesity increases mortality and recurrence rates, elevates the risk of other morbidities, and negatively impacts quality of life for BCS [292]. Obesity is also linked to various psychopathological disorders, particularly depression [293], and it has been reported that BCSs often experience cancer-related mood disturbances [294]. In addition, weight loss can help to reduce the volume of lymphedema, a common side effect in BC [295].

On this matter, a metanalysis by the Cochrane Collaboration concluded that weight loss interventions improve quality of life and anthropometrical indices in overweight women with BC, but the results on survival and recurrence are uncertain, and the research does not indicate the best type of diet. However, other reviews of the literature strongly encourage a weight loss through a healthy lifestyle and a high-quality diet, such as the MD, for ameliorating the prognosis of BC [290,291]. On the other hand, while Chen et al. reported a strong inverse association between adherence to the MD and mortality among BCS, they found no link between adherence to the MD and quality of life and anthropometric and metabolic parameters, with the exception of lower blood-glucose levels in the most adherent patients. These results might be attributed to the small sample sizes of the studies included in the systematic review and meta-analysis or the different cut-off points used for assessing the adherence to the Mediterranean diet. Therefore, more studies are needed to strengthen the evidence for all these findings [271].

Regarding the effects of lifestyle interventions on anthropometric outcomes, including body weight, BMI, body fat, or waist circumference values, a systematic review of 17 reviews found that 12 of them reported a significant reduction in at least one of these outcomes. These results were derived from multimodal programs comprising physical activity and dietary interventions, and as for the latter, the most commonly used were calorie restriction and low-fat, plant-based diets. In particular, this review highlights the importance of physical activity, especially when aerobic and resistance exercise were combined, in the promotion of psychological benefits and quality of life, while regarding biomarkers, not consistent findings were observed [291].

A significant reduction in BMI and waist circumference values was observed in a sample of BCS who underwent a six-month educational lifestyle intervention. This intervention combined dietary recommendations based on the MD model and adapted physical activity [296].

Another study, which investigated the effect of lifestyle on quality of life in a cohort of BCS, found better results among patients with a higher level of physical activity, while adherence to the MD did not appear to have significant effects [297]. The same study included a systematic review of nine studies, which investigated the role of dietary interventions on quality of life among BCS. All the studies reported a significant improvement. However, except for one study, which used exclusively a dietary intervention, all the others implemented combined programs that also included physical activity. Therefore, it is difficult to establish the specific role of diet, and more specifically, the best dietary program for these patients [297].

Besides the need for well-designed studies, another aspect that should be considered is the duration of interventions. It is well known that long-term interventions with a long follow-up are essential for the maintenance of the results over time [291].

Long and intensive behavioral interventions lasting approximately six months may be necessary to achieve and sustain behavioral changes [298].

Moreover, these interventions may need to be adapted to cultural and socioeconomic factors [298]. Some protocols aimed at preventing weight gain in the early stages of treatment have shown that higher adherence is associated with more intense personal involvement. This involvement is often implemented during hospital visits which are times of extreme fragility for patients. This helps ensure a strong adherence to the nutritional protocol, but the program may lose appeal if conducted by telephone [299].

Nonetheless, in a randomized controlled trial, a remotely delivered weight loss program that included both nutritional and physical activity counseling led to significant improvements in eating and physical activity behaviors. These were assessed using a 24 h recall for eating and an accelerometer for physical activity [300], although measuring physical activity objectively is not an easy task [301].

Finally, it has been observed that patients are often inclined to adopt and follow a healthy diet and lifestyle after undergoing chemotherapy or radiotherapy. Therefore, it would be crucial to provide them with practical and effective support to facilitate this change [302].

Parekh et al. reported that women affected by BC strongly felt that the intervention by a nutritionist was helpful in improving their cooking and eating behavior. They also noted that social support was an important aspect of the group format and found it to be helpful. In terms of changes, participants expressed a desire to continue receiving advice and resources for months after the sessions ended. Additionally, they felt that more detailed information on nutrition and physical activity would be beneficial [303].

Over time, dietary self-monitoring tends to decrease. Therefore, strategies are needed to prompt its use. The formal involvement of friends or family members in at least one initial educational session or the provision of written information that explains the rationale behind the lifestyle changes specific for BC survivors may be helpful [304].

Based on the findings discussed in this review, we believe there is a strong rationale for evaluating research protocols aimed at preventing or reducing the common weight gain experienced by BC patients during the treatment. This belief is further supported by the willingness of many patients to participate.

Such protocols should be tailored and detailed, taking into account both the individual’s caloric intake needs and the optimal composition of diet.

Considering that these could be long-term pathways, spanning years, protocols could be developed to emphasize a regular physical activity and the balanced consumption of foods, especially those rich in nutraceuticals, that have the potential to counteract disease progression and relapse. Furthermore, taking advantage of the seasonality of fruits and vegetables, they should periodically recommend the appropriate consumption of specific foods, flavorings, and spices.

The goal of these kinds of protocols would be to evaluate whether nutraceuticals can produce significant effects in the treatment of BC, either through synergy with other substances in the same food or in combination with other foods. So far, studies have been directed toward the use of these substances in the form of supplements, but it would be interesting to carefully incorporate them into a proper dietary style.

## 9. Conclusions

Growing evidence shows that nutrition can be an important complementary part in the treatment of BC. Most of the scientific literature indicates the MD as a valid and beneficial nutritional approach for BC patients in various stages of the disease, potentially improving survival and quality of life. These beneficial effects could be also attributed to the diet’s richness in nutraceutical compounds and to their synergistic activity against oxidative stress, inflammation, and cancer-related processes. However, more well- designed studies with extended follow-up are needed to establish the effects of nutritional interventions based on the Mediterranean model on weight loss among BCS patients with overweight or obesity. The support of a multidisciplinary team, including trained nutritionists, should be an integral part of patient care to reduce the risk of malnutrition and assist in weight management.



**Take-Home Messages**

Importance of the Mediterranean Diet: Our study highlights that the Mediterranean diet can have a significant impact on survival and quality of life in breast cancer patients.Role of the Gut Microbiome: Diet is a key factor in modulating the gut microbiome, which in turn can influence the efficacy of cancer therapies.Nutraceuticals and Food Synergy: Foods rich in nutraceuticals can offer additional benefits, especially when consumed in combination, due to their synergistic activity against oxidative stress and inflammation.Need for Multidisciplinary Approaches: A multidisciplinary team that includes trained nutritionists is essential for reducing the risk of malnutrition and assisting in weight management.Further Studies Needed: While the findings are promising, more well-designed studies with extended follow-up are needed to confirm these results, especially in relation to weight loss in overweight or obese patients.


## Figures and Tables

**Figure 1 antioxidants-12-01845-f001:**
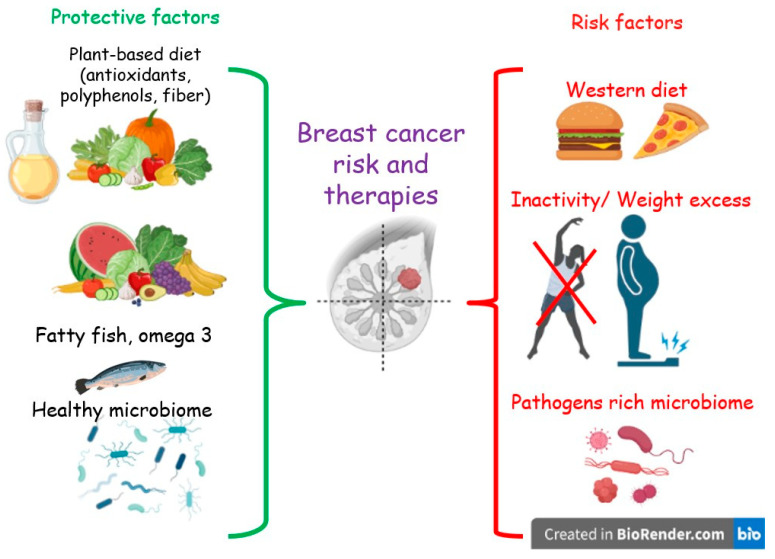
Protective and risk lifestyle factors for breast cancer.

**Figure 2 antioxidants-12-01845-f002:**
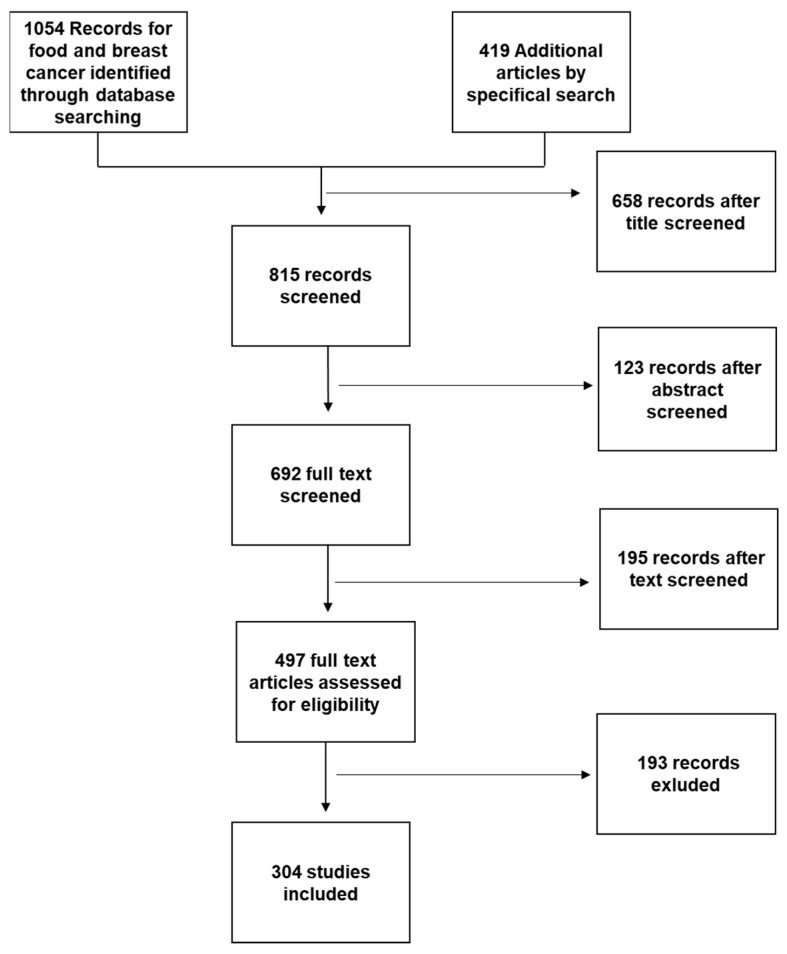
Working flowchart for study selection.

**Figure 3 antioxidants-12-01845-f003:**
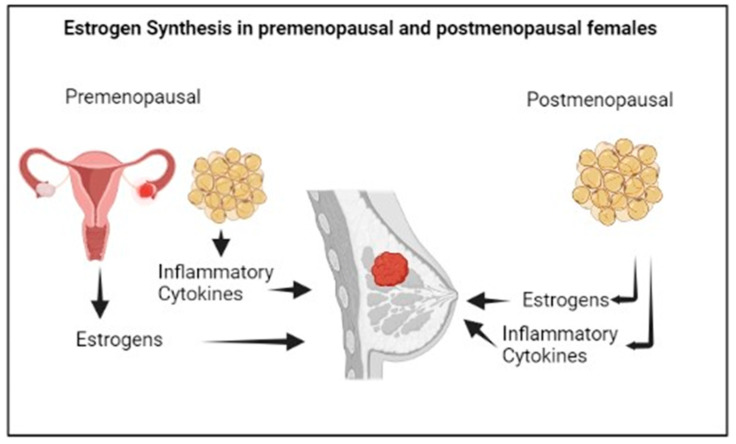
Estrogen synthesis in female sex.

**Figure 4 antioxidants-12-01845-f004:**
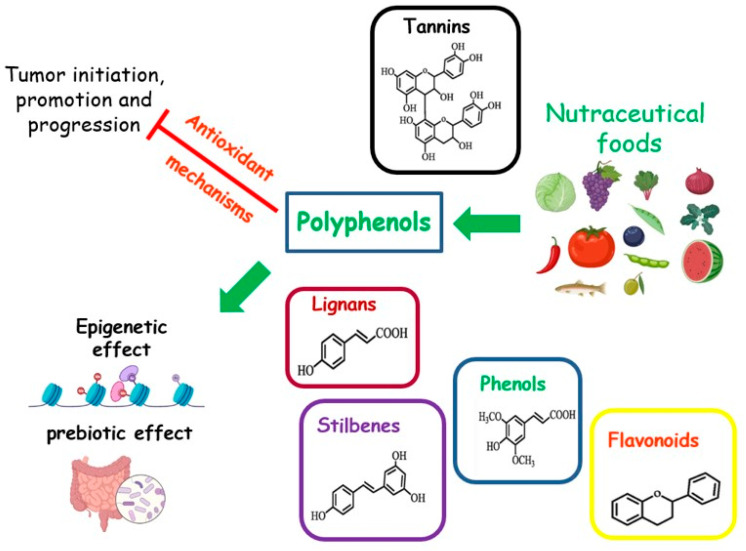
Effects of nutraceutical foods on tumor dynamics.

**Table 1 antioxidants-12-01845-t001:** Risk and protective factors before and after breast cancer diagnosis.

	Evidence	BC Risk before Diagnosis		BC Risk (Survivors)
		Premenopausal	Postmenopausal	
**Reduced risk**	Probable	Vigorous physical activity Body fatness Lactation	Physical activity Body fatness in young adulthood Lactation	
	Suggestive/ limited	Non-starchy vegetables (ER breast cancers only) Dairy products Foods containing carotenoids Diets high in calcium Physical activity	Non-starchy vegetables (ER breast cancers only) Foods containing carotenoids Diets high in calcium	Physical activity Fiber-containing foods Soy foods
**Increased risk**	Convincing	Adult attained height *	Alcoholic drinks Body fatness Adults weight gain Adult attained height *	
	Probable	Alcoholic drinks Greater birthweight		
	Suggestive/ limited			Body fat # § ç Saturated Fatty Acids #

Table legend: Time of exposure: #, before diagnosis; §, less than one year after diagnosis; ç, one year or more after diagnosis. ER, estrogen receptor. * According to the WCRF reports, it is unlikely that the adult attained height has any influence on cancer risk. It is probably a marker of genetic, environmental, hormonal, and nutritional factors that influence growth in the early years of life.

## Data Availability

Data are available from the authors upon reasonable request.

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
