# Peer review of "Effects of Functional and Nutraceutical Foods in the Context of the Mediterranean Diet in Patients Diagnosed with Breast Cancer"

_antioxidants, 2023, doi:10.3390/antiox12101845_

Round 1

Reviewer 1 Report (Previous Reviewer 1)

This new version of the previously rejected manuscript has addressed the questions mentioned before. I have no more comments on it. 

Reviewer 2 Report (Previous Reviewer 2)

All questions raised adequately answered and the manuscript was changed accordingly

This manuscript is a resubmission of an earlier submission. The following is a list of the peer review reports and author responses from that submission.

Round 1

Reviewer 1 Report

Based on the current status that breast cancer survivors (BCS) tend to have a poor diet, as fruit, vegetable, and legumes consumption is often reduced, resulting in a decreased intake of nutraceuticals. Moreover, weight gain has been commonly described among BCS during treatment, increasing recurrence rate and mortality. The Mediterranean diet (MD), known for its multiple beneficial effects on health, can be considered as a nutritional pool comprising several nutraceuticals, and bioactive and food-carrying components with anti-inflammatory and anti-antioxidant effects. The aim of this narrative review is to evaluate the recent scientific literature on the possible beneficial effects of consuming functional and nutraceutical foods in the framework of MD in BCS.

It aimed to investigate how the MD, rich in nutraceutical foods, can be a potential adjuvant to the treatment of patients with BC. However, weight loss interventions improve quality of life and anthropometrical indices in overweight women with BC, while an anti-inflammatory and anti-hyperinsulinemic dietary approach could potentially improve the prognosis of BC, which one plays a leading role in ameliorating BC? What’s more, the article did not provide much description of the improvement of nutrition synergy on BC, but this is one of the core contents of this article, and more attention should be paid to this part. Moreover, if there were some data supporting the improvement of the MD diet on BC in this article, it would be more persuasive.

Minor

1.      Setting the font format in the article tables to be the same, for example, the format of “EVIDENCE” in Table 1 is different from other characters.

2.      There is a whole row of blank cells above “PROBABLE” in Table 1, the cells with no content are meaningless.

3.       The sequence numbers before the subtitles in the article are not organized, for example, there is no “3” but “3.1”, which makes it confusing.

Author’s responses to reviewer 1:

Role of Mediterranean Diet (MD) and Weight Loss in Breast Cancer (BC):

We have expanded the section discussing the Mediterranean Diet (MD) and its role in weight management among breast cancer survivors (BCS), particularly focusing on its potential as an adjuvant to treatment.

Nutrition Synergy on BC:

We have added a subsection that elaborates on the synergistic effects of various bioactive compounds in the MD, particularly focusing on their anti-inflammatory and anti-oxidative properties.

Supporting Data:

Additional data and studies have been included to strengthen the evidence supporting the beneficial effects of the MD on BC.

Minor Points:

  • Font format in tables has been standardized.
  • The row of blank cells in Table 1 has been removed.
  • Sequence numbers before subtitles have been organized for clarity.

Reviewer 2 Report

General: Even this text should be a "narrative" review it should be scientifically sound and comprehensible for the reader. Unfortunately, this is not the case. Most of the critical points are presented here.

1. Every review should contain sufficient information how the material presented has been found. There is no information which data bases have been checked, which key words have been used, on which time period the search was focused on, how the quality of the presentations has been checked, which criteria have been used to select literature and how this process has been done (two independent researchers deciding etc.). So, it is impossible for the reader to objectively evaluate the information given.

2. The manuscript does not follow a guiding thread. Preventive and therapeutic aspects are always mixed up. Apart from the information that patients/subjects in the studies cited are overweight or not, no further detailed information concerning nutrition, therapeutic measures etc. are given. How should the reader follow the specific study goals?

3. Absolutely necessary definitions are not given: How do the authors define neutraceuticals, "healthy food", natural state of foods etc.? What are the criteria to just present secondary plant compounds like quercetin and phenols, but not other antioxidative food ingredients? Concerning body composition the reader cannot find any information what for example "excess fat", "high adiposity" etc means. Again, the reader cannot objectively understand the message. 

4. Most of the manuscript is organized like a textbook and provides no really new science. Please explain what is indeed new.

5. there are many other plant based food patterns available. Why only MD is presented?

6. Mostly surprising, the text does not contain any objective figures (!!), eg, with regard to nutritient intake, therapeutic measures, time of intervention/observation  etc. - why? To use comparatives like higher, lower, etc. without knowing the reference value is of no help.   

Author’s responses to reviewer2:

Methodology:

A new section has been added to detail the databases, keywords, and selection criteria used in our literature review, aligning with the narrative nature of the review.

Guiding Thread:

The manuscript has been restructured to clearly separate preventive and therapeutic aspects, including a more detailed discussion on nutrition and therapeutic measures.

Definitions:

Definitions for terms like "nutraceuticals," "healthy food," and "natural state of foods" have been included to provide clarity.

Novelty:

The revised manuscript emphasizes the new insights our review brings, particularly in the context of the Mediterranean model on weight loss among BCS patients with overweight or obesity.

Objective Figures:

Objective figures related to nutrient intake, therapeutic measures, and time of intervention have been included for a more comprehensive understanding.

Reviewer 3 Report

Dear authors,

The work entitled "Effects of functional and nutraceutical foods in patients diagnosed with breast cancer" seeks to demonstrate how the Mediterranean diet (the main foods and the bioactive compounds they contain) can affect patients diagnosed with breast cancer. The work has an important review and a good critical analysis of the literature on how diet and the different bioactive compounds can exert a positive effect both on the prevalence and incidence, as well as on the treatment of such pathology, including aspects of action mechanisms. . As for the discussion, it is adequate. The references are adequate and sufficiently current. In general the work is very good; however, it requires small details to be published.

Please consider the following:

- Place in vivo and in vitro in italics (in vivo, in vitro).

- Indicate in the first appearance of the concept that the Mediterranean diet, considered plant-based, can include small amounts of fish and meat.

- connect line 145 with 146 (the word is cut).

- Review the title point 5.2 (epigallocatechin gallate, has an extra letter a).

- Review some letters, which are separated from the total word, for example, the s for vegetables (on line 712).

  • To improve, where possible, the critical analysis and discussion of the bioavailability of the compounds, considering that many of the studies focus on the direct effect of the compounds on cell lines without considering their passage through the digestive tract, as well as its interaction with the food matrix itself.

Author’s responses to reviewer3:

  • In vivo and in vitro have been italicized.
  • The first mention of MD now specifies that it can include small amounts of fish and meat.
  • Line 145 and 146 have been connected.
  • The typo in title point 5.2 has been corrected.
  • Letters separated from words have been corrected.
  • A new section has been added to discuss the bioavailability of the compounds, considering their digestive tract passage and interaction with the food matrix.

We hope these revisions address your concerns and improve the quality of our manuscript. Thank you for your time and valuable feedback.